# Mildly Constrained Evaluation Policy for Offline Reinforcement Learning

## Abstract

Offline reinforcement learning (RL) methodologies enforce constraints on the policy to adhere closely to the behavior policy, thereby stabilizing value learning and mitigating the selection of out-of-distribution (OOD) actions during test time. Conventional approaches apply identical constraints for both value learning and test time inference. However, our findings indicate that the constraints suitable for value estimation may in fact be excessively restrictive for action selection during test time. To address this issue, we propose a *Mildly Constrained Evaluation Policy (MCEP)* for test time inference with a more constrained *target policy* for value estimation. Since the *target policy* has been adopted in various prior approaches, MCEP can be seamlessly integrated with them as a plug-in. We instantiate MCEP based on TD3-BC [Fujimoto and Gu, 2021] and AWAC [Nair et al., 2020] algorithms. The empirical results on MuJoCo locomotion tasks show that the MCEP significantly outperforms the *target policy* and achieves competitive results to state-of-the-art offline RL methods. The codes are open-sourced at *link*.

## 1 Introduction

Offline reinforcement learning (RL) extracts a policy from data that is pre-collected by unknown policies. This setting does not require interactions with the environment thus it is well-suited for tasks where the interaction is costly or risky. Recently, it has been applied to Natural Language Processing [Snell et al., 2022], e-commerce [Degirmenci and Jones] and real-world robotics [Kalashnikov et al., 2021, Rafailov et al., 2021, Kumar et al., 2022, Shah et al., 2022] etc. Compared to the standard online setting where the policy gets improved via trial and error, learning with a static offline dataset raises novel challenges. One challenge is the distributional shift between the training data and the data encountered during deployment. To attain stable evaluation performance under the distributional shift, the policy is expected to stay close to the behavior policy. Another challenge is the "extrapolation error" [Fujimoto et al., 2019, Kumar et al., 2019] that indicates value estimate error on unseen state-action pairs or Out-Of-Distribution (OOD) actions. Worsely, this error can be amplified with bootstrapping and cause instability of the training, which is also known as deadly-triad [Van Hasselt et al., 2018]. Majorities of model-free approaches tackle these challenges by either constraining the policy to adhere closely to the behavior policy [Wu et al., 2019, Kumar et al., 2019, Fujimoto and Gu, 2021] or regularising the Q to pessimistic estimation for OOD actions [Kumar et al., 2020, Lyu et al., 2022]. In this work, we focus on *policy constraints* methods.

Policy constraints methods minimize the disparity between the policy distribution and the behavior distribution. It is found that policy constraints introduce a tradeoff between stabilizing value estimates and attaining better performance. While previous approaches focus on developing various constraints for the learning policy to address this tradeoff, the tradeoff itself is not well understood. Current solutions have confirmed that an excessively constrained policy enables stable values estimate

but degrades the evaluation performance [Kumar et al., 2019, Singh et al., 2022, Yu et al., 2023]. Nevertheless, it is not clear to what extent this constraint fails to stabilize value learning and to what extent this constraint leads to a performant evaluation policy. It is essential to investigate these questions as their answers indicate how well a solution can be found under the tradeoff. However, the investigation into the latter question is impeded by the existing tradeoff, as it requires tuning the constraint without influencing the value learning. We circumvent the tradeoff and seek solutions for this investigation through the critic. For actor-critic methods, [Czarnecki et al., 2019] has shed light on the potential of distilling a student policy that improves over the teacher using the teacher's critic. Inspired by this work, we propose to derive an extra *evaluation policy* from the critic to avoid solving the above-mentioned tradeoff. The actor is now called *target policy* as it is used only to stabilize the value estimation.

Based on the proposed framework, we empirically investigate the constraint strengths for 1) stabilizing value learning and 2) better evaluation performance. The results find that a milder constraint improves the evaluation performance but may fall beyond the constraint space of stable value estimation. This finding indicates that the optimal evaluation performance may not be found under the tradeoff, especially when stable value learning is the priority. Consequently, we propose a novel approach of using a *Mildly Constrained Evaluation Policy (MCEP)* derived from the critic to avoid solving the above-mentioned tradeoff and to achieve better evaluation performance.

As the *target policy* is commonly used in previous approaches, our MCEP can be integrated with them seamlessly. In this paper, we first validate the finding of [Czarnecki et al., 2019] in the offline setting by a toy maze experiment, where a constrained policy results in bad evaluation performance but its off-policy Q estimation indicates an optimal policy. After that, our experiments on D4RL [Fu et al., 2020] MoJoCo locomotion tasks showed that in most tasks milder constraint achieves better evaluation performance while more restrictive constraint stabilizes the value estimate. Finally, we instantiated MCEP on both TD3BC and AWAC algorithms. The empirical results find that the MCEP significantly outperforms the *target policy* and achieves competitive results to state-of-the-art offline RL methods.

## 2   Related Work

Policy constraints method (or behavior-regularized policy method) [Wu et al., 2019, Kumar et al., 2019, Siegel et al., 2020, Fujimoto and Gu, 2021] forces the policy distribution to stay close to the behavior distribution. Different discrepancy measurements such as KL divergence [Jaques et al., 2019, Wu et al., 2019], reverse KL divergence Cai et al. [2022] and Maximum Mean Discrepancy [Kumar et al., 2019] are applied in previous approaches. [Fujimoto and Gu, 2021] simply adds a behavior-cloning (BC) term to the online RL method Twin Delayed DDPG (TD3) [Fujimoto et al., 2018] and obtains competitive performances in the offline setting. While the above-mentioned methods calculate the divergence from the data, [Wu et al., 2022] estimates the density of the behavior distribution using VAE, and thus the divergence can be directly calculated. Except for explicit policy constraints, implicit constraints are achieved by different approaches. E.g. [Zhou et al., 2021] ensures the output actions stay in support of the data distribution by using a pre-trained conditional VAE (CVAE) decoder that maps latent actions to the behavior distribution. In all previous approaches, the constraints are applied to the learning policy that is queried during policy evaluation and is evaluated in the environment during deployment. Our approach does not count on this learning policy for the deployment, instead, it is used as a *target policy* only for the policy evaluation.

While it is well-known that a policy constraint can be efficient to reduce extrapolation errors, its drawback is not well-studied yet. [Kumar et al., 2019] reveals a tradeoff between reducing errors in the Q estimate and reducing the suboptimality bias that degrades the evaluation policy. A constraint is designed to create a policy space that ensures the resulting policy is under the support of the behavior distribution for mitigating bootstrapping error. [Singh et al., 2022] discussed the inefficiency of policy constraints on *heteroskedastic* dataset where the behavior varies across the state space in a highly non-uniform manner, as the constraint is state-agnostic. A reweighting method is proposed to achieve a state-aware distributional constraint to overcome this problem. Our work studies essential questions about the tradeoff [Kumar et al., 2019] and overcomes this drawback [Singh et al., 2022] by using an extra evaluation policy.

There are methods that extract an evaluation policy from a learned Q estimate. One-step RL [Brand-fonbrener et al., 2021] first estimates the behavior policy and its Q estimate, which is later used for extracting the evaluation policy. Although its simplicity, one-step RL is found to perform badly in long-horizon problems due to a lack of iterative dynamic programming [Kostrikov et al., 2022]. [Kostrikov et al., 2022] proposed Implicity Q learning (IQL) that avoids query of OOD actions by learning an upper expectile of the state value distribution. No explicit target policy is modeled during their Q learning. With the learned Q estimate, an evaluation policy is extracted using advantage-weighted regression [Wang et al., 2018, Peng et al., 2019]. Our approach has a similar form of extracting an evaluation from a learned Q estimate. However, one-step RL aims to avoid distribution shift and iterative error exploitation during iterative dynamic programming. IQL avoids error exploitation by eliminating OOD action queries and abandoning policy improvement (i.e. the policy is not trained against the Q estimate). Our work instead tries to address the error exploitation problem and evaluation performance by using policies of different constraint strengths.

## 3 Background

We model the environment as a Markov Decision Process (MDP) $\langle S, A, R, T, p_0(s), \gamma, \rangle$, where $S$ is the state space, $A$ is the action space, $R$ is the reward function, $T(s'|s, a)$ is the transition probability, $p_0(s)$ is initial state distribution and $\gamma$ is a discount factor. In the offline setting, a static dataset $\mathcal{D}_\beta = \{(s, a, r, s')\}$ is pre-collected by a behavior policy $\pi_\beta$. The goal is to learn a policy $\pi_\phi(s)$ with the dataset $\mathcal{D}$ that maximizes the discounted cumulated rewards in the MDP:

$$\phi^* = \arg\max_\phi \mathbb{E}_{s_0 \sim p_0(\cdot), a_t \sim \pi_\phi(s_t), s_{t+1} \sim T(\cdot|s_t, a_t)} [\sum_{t=0}^{\infty} \gamma^t R(s_t, a_t)] \tag{1}$$

Next, we introduce the general policy constraint method, where the policy $\pi_\phi$ and an off-policy Q estimate $Q_\theta$ are updated by iteratively taking policy improvement steps and policy evaluation steps, respectively. The policy evaluation step minimizes the Bellman error:

$$\mathcal{L}_Q(\theta) = \mathbb{E}_{s_t, a_t \sim \mathcal{D}, a_{t+1} \sim \pi_\phi(s_{t+1})} [(Q_\theta(s_t, a_t) - (r + \gamma Q_{\theta'}(s_t, a_{t+1})))^2]. \tag{2}$$

where the $\theta'$ is the parameter for a delayed-updated target Q network. The Q value for the next state is calculated with actions $a_{t+1}$ from the learning policy that is updated through the policy improvement step:

$$\mathcal{L}_\pi(\phi) = \mathbb{E}_{s \sim \mathcal{D}, a \sim \pi_\phi(s)} [-Q_\theta(s, a) + wC(\pi_\beta, \pi_\phi)], \tag{3}$$

where $C$ is a constraint measuring the discrepancy between the policy distribution $\pi_\phi$ and the behavior distribution $\pi_\beta$. The $w \in (0, \infty]$ is a weighting factor. Different kinds of constraints were used such as Maximum Mean Discrepancy (MMD), KL divergence, and reverse KL divergence.

## 4 Method

In this section, we first introduce the generic algorithm that can be integrated with any policy constraints method. Next, we introduce two examples based on popular offline RL methods TD3BC and AWAC. With a mildly constrained evaluation policy, we name these two instances as *TD3BC with MCEP (TD3BC-MCEP)* and *AWAC with MCEP (AWAC-MCEP)*.

### 4.1 Offline RL with mildly constrained evaluation policy

The proposed method is designed for overcoming the tradeoff between a stable policy evaluation and a performant evaluation policy. In previous constrained policy methods, a restrictive policy constraint is applied to obtain stable policy evaluation. We retain this benefit but use this policy (actor) $\tilde{\pi}$ as a *target policy* only to obtain stable policy evaluation. To achieve better evaluation performance, we introduce an MCEP $\pi^e$ that is updated by taking policy improvement steps with the critic $Q_{\tilde{\pi}}$. Different from $\tilde{\pi}$, $\pi^e$ does not participate in the policy evaluation procedure. Therefore, a mild policy constraint can be applied, which helps $\pi^e$ go further away from the behavior distribution without influencing the stability of policy evaluation. We demonstrate the policy spaces and policy trajectories for $\tilde{\pi}$ and $\pi^e$ in the l.h.s. diagram of Figure 1, where $\pi^e$ is updated in the wider policy space using $Q_{\tilde{\pi}}$.

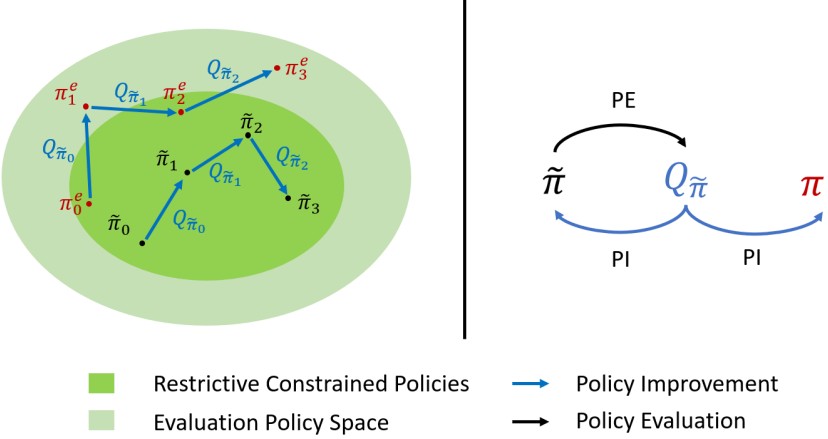

| Restrictive Constrained Policies | ⟶ (blue) | Policy Improvement |
| Evaluation Policy Space | ⟶ (black) | Policy Evaluation |

Figure 1: **Left:** diagram depicts policy trajectories for target policy $\tilde{\pi}$ and MCEP $\pi^e$. **Right:** policy evaluation steps to update $Q_{\tilde{\pi}}$ and policy improvement steps to update $\tilde{\pi}$ and $\pi^e$.

The overall algorithm is shown as pseudo-codes (Alg. 1). At each step, the $Q_{\tilde{\pi}}$, $\tilde{\pi}_\psi$ and $\pi_\phi^e$ are updated iteratively. A policy evaluation step updates $Q_{\tilde{\pi}}$ by minimizing the TD error (line 7), i.e. the deviation between the approximate $Q$ and its target value. Next, a policy improvement step updates $\tilde{\pi}_\psi$ (line 6. These two steps form the actor-critic algorithm. After that, $\pi_\phi^e$ is extracted from the $Q_{\tilde{\pi}}$, by taking a policy improvement step with a policy constraint that is likely milder than the constraint for $\tilde{\pi}_\psi$ (line 7). Many approaches can be taken to obtain a milder

---

**Algorithm 1** MCEP Training

1: **Hyperparameters:** LR $\alpha$, EMA $\eta$, $\tilde{w}$ and $w^e$
2: **Initialize:** $\theta, \theta', \psi$, and $\phi$
3: **for** i=1, 2, ..., N **do**
4: $\quad \theta \leftarrow \theta - \alpha \mathcal{L}_Q(\theta)$ (Equation 2)
5: $\quad \theta' \leftarrow (1 - \eta)\theta' + \eta\theta$
6: $\quad \psi \leftarrow \psi - \alpha \mathcal{L}_{\tilde{\pi}}(\psi; \tilde{w})$ (Equation 3)
7: $\quad \phi \leftarrow \phi - \alpha \mathcal{L}_{\pi^e}(\phi; w^e)$ (Equation 3)

---

policy constraint. For example, tuning down the weight factor $w^e$ for the policy constraint term or replacing the constraint measurement with a less restrictive one. Note that the constraint for $\pi_\phi^e$ is necessary (the constraint term should not be dropped) as the $Q_{\tilde{\pi}}$ has large approximate errors for state-action pairs that are far from the data distribution.

### 4.2 Two Examples: TD3BC-MCEP and AWAC-MCEP

**TD3BC with MCEP** TD3BC takes a minimalist modification on the online RL algorithm TD3. To keep the learned policy to stay close to the behavior distribution, a behavior-cloning term is added to the policy improvement objective. TD3 learns a deterministic policy therefore the behavior cloning is achieved by directly regressing the data actions. For TD3BC-MCEP, the *target policy* $\tilde{\pi}_\psi$ has the same policy improvement objective as TD3BC:

$$\mathcal{L}_{\tilde{\pi}}(\psi) = \mathbb{E}_{(s,a)\sim\mathcal{D}}[-\tilde{\lambda}Q_\theta(s, \tilde{\pi}_\psi(s)) + (a - \tilde{\pi}_\psi(s))^2], \tag{4}$$

where the $\tilde{\lambda} = \frac{\tilde{\alpha}}{\frac{1}{N}\sum_{s_i,a_i}|Q_\theta(s_i,a_i)|}$ is a normalizer for Q values with a hyper-parameter $\tilde{\alpha}$: The $Q_\theta$ is updated with the policy evaluation step similar to Eq. 2 using $\tilde{\pi}_\psi$. The MCEP $\pi_\phi^e$ is updated by policy improvement steps with the $Q_{\tilde{\pi}}$ taking part in. The policy improvement objective function for $\pi_\phi^e$ is similar to Eq. 4 but with a higher-value $\alpha^e$ for the Q-value normalizer $\lambda^e$. The final objective for $\pi_\phi^e$ is

$$\mathcal{L}_{\pi^e}(\phi) = \mathbb{E}_{(s,a)\sim\mathcal{D}}[-\lambda^e Q(s, \pi_\phi^e(s)) + (a - \pi_\phi^e(s))^2]. \tag{5}$$

**AWAC with MCEP** AWAC [Nair et al., 2020] is an advantage-weighted behavior cloning method. As the target policy imitates the actions from the behavior distribution, it stays close to the behavior distribution during learning. In AWAC-MCEP, the policy evaluation follows the Eq. 2 with the target

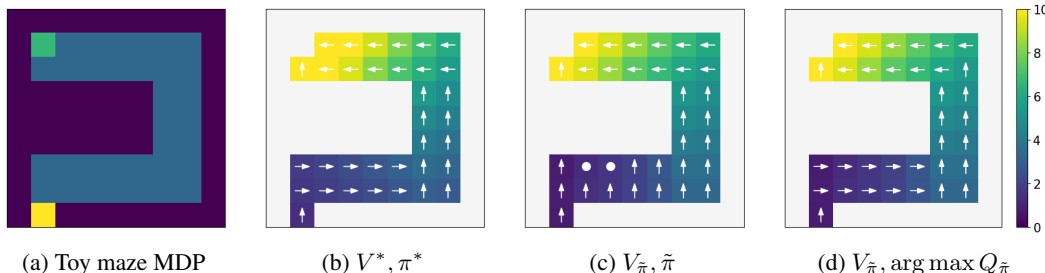

| (a) Toy maze MDP | (b) $V^*, \pi^*$ | (c) $V_{\tilde{\pi}}, \tilde{\pi}$ | (d) $V_{\tilde{\pi}}, \arg\max Q_{\tilde{\pi}}$ |

Figure 2: Evaluation of policy constraint method on a toy maze MDP 2a. In other figures, the color of a grid represents the state value and arrows indicate the actions from the corresponding policy. 2b shows the optimal value function and one optimal policy. 2c shows a constrained policy trained from the above-mentioned offline data, with its value function calculated by $V_\pi = \mathbb{E}_a Q(s, \pi(a|s))$. The policy does not perform well in the low state-value area but its value function is close to the optimal value function. 2d indicates that an optimal policy is recovered by deriving the greedy policy from the off-policy Q estimate (the critic).

163     policy $\tilde{\pi}_\psi$ that updates with the following objective:

$$\mathcal{L}_{\tilde{\pi}}(\psi) = \mathbb{E}_{s,a \sim \mathcal{D}}\left[-\exp\left(\frac{1}{\tilde{\lambda}}A(s,a)\right)\log\tilde{\pi}_\psi(a|s)\right], \tag{6}$$

164 where the advantage $A(s,a) = Q_\theta(s,a) - Q_\theta(s, \tilde{\pi}_\psi(s))$. This objective function solves an advantage-
165 weighted maximum likelihood. Note that the gradient will not be passed through the advantage term.
166 As this objective has no policy improvement term, we use the original policy improvement with KL
167 divergence as the policy constraint and construct the following policy improvement objective:

$$\mathcal{L}_{\pi^e}(\phi) = \mathbb{E}_{s,a \sim \mathcal{D}, \hat{a} \sim \pi^e(\cdot|s)}[-Q(s,\hat{a}) + \lambda^e D_{KL}\big(\pi_\beta(\cdot|s)||\pi_\phi^e(\cdot|s)\big)] \tag{7}$$

$$= \mathbb{E}_{s,a \sim \mathcal{D}, \hat{a} \sim \pi^e(\cdot|s)}[-Q(s,\hat{a}) - \lambda^e \log \pi_\phi^e(a|s)], \tag{8}$$

168 where the weighting factor $\lambda^e$ is a hyper-parameter. Although the Eq. 6 is derived by solving Eq. 8
169 in a parametric-policy space, the original problem (Eq. 8) is less restrictive even with $\tilde{\lambda} = \lambda^e$ as it
170 includes a $-Q(s, \pi^e(s))$ term. This difference means that even with a $\lambda^e > \tilde{\lambda}$, the policy constraint
171 for $\pi^e$ could still be more relaxed than the policy constraint for $\tilde{\pi}$.

## 5   Experiments

173 In this section, we set up 4 groups of experiments to illustrate: 1) the policy constraint might degrade
174 the evaluation performance by forcing the policy to stay close to low-state-value transitions. 2) The
175 suitable constraint for the final inference could be milder than the ones for safe Q estimates. 3) Our
176 method shows significant performance improvement compared to the target policy and achieves
177 competitive results to state-of-the-art offline RL methods on MuJoCo locomotion tasks. 4) the MCEP
178 generally gains a higher estimate Q compared to the target policy. Additionally, we adopt 2 groups of
179 ablation studies to verify the benefit of an MCEP and to investigate the constraint strengths of MCEP.

180 **Environments** D4RL [Fu et al., 2020] is an offline RL benchmark consisting of many task sets.
181 Our experiments involve MuJoCo locomotion tasks (*-v2*) and two tasks from Adroit (*-v0*). For
182 MuJoCo locomotion tasks, we select 4 versions of datasets: data collected by a uniformly-random
183 agent (*random*), collected by a medium-performance policy (*medium*), a $50\% - 50\%$ mixture of the
184 medium data and the replay buffer during training a medium-performance policy (*medium-replay*), a
185 $50\% - 50\%$ mixture of the medium data and expert demonstrations (*medium-expert*). For Adroit,
186 we select *pen-human* and *pen-cloned*, where the *pen-human* includes a small number of human
187 demonstrations, and *pen-cloned* is a $50\% - 50\%$ mixture of demonstrations and data collected by
188 rolling out an imitation policy on the demonstrations.

### 5.1   Target policy that enables safe Q estimate might be overly constrained

190 To investigate the policy constraint under a highly suboptimal dataset, we set up a toy maze MDP that
191 is similar to the one used in [Kostrikov et al., 2022]. The environment is depicted in Figure 2a, where

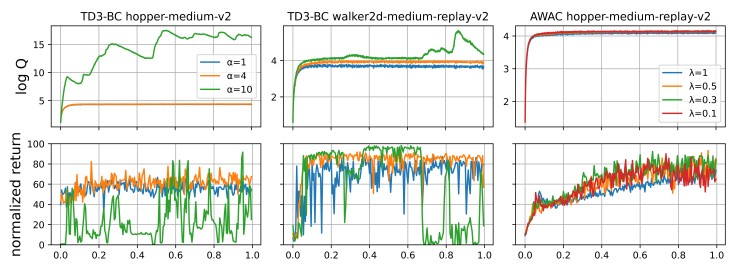

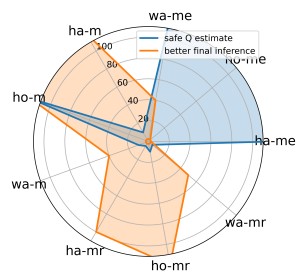

Figure 4: The training process of TD3BC and AWAC. **Left:** TD3BC on *hopper-medium-v2*. **Middle:** TD3BC on *walker2d-medium-replay-v2*. **Right:** AWAC on *hopper-medium-replay-v2*.

Figure 5: $\alpha$ values in TD3BC for value estimate and test time inference in MuJoCo locomotion tasks.

the lower left yellow grid is the starting point and the upper left green grid is the terminal state that gives a reward of 10. Other grids give no reward. Dark blue indicates un-walkable areas. The action space is defined as 4 direction movements (arrows) and staying where the agent is (filled circles). There is a 25% probability that a random action is taken instead of the action from the agent. For the dataset, 99 trajectories are collected by a uniformly random agent and 1 trajectory is collected by an expert policy. Fig. 2b shows the optimal value function (colors) and one of the optimal policies.

We trained a constrained policy using Eq. 2 and Eq. 8 in an actor-critic manner, where the actor is constrained by a KL divergence with a weight factor of 1. Figure 2c shows the value function and the policy. We observe that the learned value function is close to the optimal one in Figure 2b. However, the policy does not make optimal actions in the lower left areas where the state values are relatively low. As the policy improvement objective shows a trade-off between the Q and the KL divergence, when the Q value is low, the KL divergence term will obtain higher priority. i.e. in low-Q-value areas, the KL divergence takes the majority for the learning objective, which makes the policy stays closer to the transitions in low-value areas. However, we find that the corresponding value function indicates an optimal policy. In Figure 2d, we recover a greedy policy underlying the learned critic that shows an optimal policy. In conclusion, the constraint might degrade the evaluation performance although the learned critic may indicate a better policy. Although such a trade-off between the Q term and the KL divergence term can be alleviated in previous work [Fujimoto and Gu, 2021] by normalizing the Q values, in the next section, we will illustrate that the constraint required to obtain performant evaluation policy can still cause unstable value estimate.

## 5.2 Test-time inference requires milder constraints

The previous experiment shows that a restrictive constraint might harm the test-time inference, which motivates us to investigate what constraints make better evaluation performance. Firstly, we relax the policy constraint on TD3BC and AWAC by setting up different hyper-parameter values that control the strengths of the policy constraints. For TD3BC, we set $\alpha = \{1, 4, 10\}$ ([Fujimoto and Gu, 2021] recommends $\alpha = 2.5$). For AWAC, we set $\lambda = \{1.0, 0.5, 0.3, 0.1\}$ ([Nair et al., 2020] recommends $\lambda = 1$). Finally, We visualize the evaluation performance and the learned Q estimates.

In Figure 4, the left two columns show the training of TD3BC in the *hopper-medium-v2* and *walker2d-medium-replay-v2*. In both domains, we found that using a milder constraint by tuning the $\alpha$ from 1 to 4 improves the evaluation performance, which motivates us to expect better performance with $\alpha = 10$. As shown in the lower row, we do observe higher performances in some training steps. However, unstable training is caused by the divergence in value estimate, which indicates the tradeoff between the stable Q estimate and the evaluation performance. The rightmost column shows the training of AWAC in *hopper-medium-replay-v2*, we observe higher evaluation performance by relaxing the constraint ($\lambda > 1$). Although the Q estimate keeps stable during the training in all $\lambda$ values, higher $\lambda$ results in unstable policy performance and causes the performance crash with $\lambda = 0.1$.

Concluding on all these examples, a milder constraint can potentially improve the performance but may cause unstable Q estimates or unstable policy performances. As we find that relaxing the constraint on current methods triggers unstable training, which hinders the investigation of constraints

| Task Name | BC | CQL | IQL | TD3BC | TD3BC-MCEP (ours) | AWAC | AWAC-MCEP (ours) |
|---|---|---|---|---|---|---|---|
| halfcheetah-r | 2.2±0.0 | - | 10±1.7 | 11.7±0.4 | 28.8±1.0 | 9.6±0.4 | **34.9±0.8** |
| hopper-r | 4.7±0.1 | - | 8.1±0.4 | 8.3±0.1 | 8.0±0.4 | 5.3±0.4 | **9.8±0.5** |
| walker2d-r | 1.6±0.0 | - | **5.6±0.1** | 1.2±0.0 | -0.2±0.1 | 5.2±1.0 | 3.1±0.4 |
| halfcheetah-m | 42.4±0.1 | 44.0 | 47.4±0.1 | 48.7±0.2 | **55.5±0.4** | 45.1±0 | 46.6±0 |
| hopper-m | 54.1±1.1 | 58.5 | 65±3.6 | 56.1±1.2 | **91.8±0.9** | 58.9±1.9 | **91.1±1.5** |
| walker2d-m | 71±1.7 | 72.5 | 80.4±1.7 | 85.2±0.9 | **88.8±0.5** | 79.6±1.5 | 83.4±0.9 |
| halfcheetah-m-r | 37.8±1.1 | 45.5 | 43.2±0.8 | 44.8±0.3 | **50.6±0.2** | 43.3±0.1 | 44.9±0.1 |
| hopper-m-r | 22.5±3.0 | 95.0 | 74.2±5.3 | 55.2±10.8 | **100.9±0.4** | 64.8±6.2 | **101.4±0.2** |
| walker2d-m-r | 14.4±2.7 | 77.2 | 62.7±1.9 | 50.9±16.1 | **86.3±3.2** | 84.1±0.6 | 84.6±1.3 |
| halfcheetah-m-e | 62.3±1.5 | **91.6** | **91.2±1.0** | 87.1±1.4 | 71.5±3.7 | 77.6±2.6 | 76.2±5.5 |
| hopper-m-e | 52.5±1.4 | 105.4 | **110.2±0.3** | 91.7±10.5 | 80.1±12.7 | 52.4±8.7 | 92.5±8.3 |
| walker2d-m-e | 107±1.1 | 108.8 | **111.1±0.5** | 110.4±0.5 | **111.7±0.3** | 109.5±0.2 | 110.3±0.1 |
| Average | 39.3 | - | 59.0 | 54.2 | 64.5 | 52.9 | **64.9** |
| pen-human | **76.8±4.8** | 37.5 | 64.2±10.4 | 61.6±11 | 58.6±20.8 | 34.7±11.8 | 23.3 ±5.6 |
| pen-cloned | 28.5±6.7 | 39.2 | 32.1±7.5 | **49±9.5** | 43.4±20.3 | 20.8±7.3 | 19.0±7.5 |
| Average | 52.6 | 38.3 | 48.1 | **55.3** | 51.0 | 27.7 | 21.1 |

Table 1: Normalized episode returns on D4RL benchmark. The results (except for CQL) are means and standard errors from the last step of 5 runs using different random seeds. Performances that are higher than corresponding baselines are underlined and task-wise best performances are bolded.

for better evaluation performance. We instead systematically study the constraint strengths in TD3BC and TD3BC with *evaluation policy* (TD3BC-EP).

We first tune the $\alpha$ for TD3BC to unveil the range for safe Q estimates. Then in TD3BC-EP, we tune the $\alpha^e$ for the evaluation policy with a fixed $\tilde{\alpha} = 2.5$ to approximate the constraint range of better test inference performance (i.e. where the evaluation policy outperforms the target policy). The $\tilde{\alpha} = 2.5$ is selected to ensure a stable Q estimate (also the paper-recommended value). The $\alpha$ ($\alpha^e$) is tuned within $\{2.5, 5, 10, 20, 30, 40, 50, 60, 70, 80, 90, 100\}$. For each $\alpha$ ($\alpha^e$), we observe the training of 5 runs with different random seeds. In Figure 5, we visualize these two ranges for each task from MuJoCo locomotion set. The blue area shows $\alpha$ values where the TD3BC Q estimate is stable for all seeds. The edge shows the lowest $\alpha$ value that causes Q value explosion. The orange area shows the range of $\alpha^e$ where the learned evaluation policy outperforms the target policy. Its edge (the orange line) shows the lowest $\alpha^e$ values where the evaluation policy performance is worse than the target policy. For each task, the orange area has a lower bound $\alpha^e = 2.5$ where the evaluation policy shows a similar performance to the target policy.

Note that $\alpha$ weighs the Q term and thus a larger $\alpha$ indicates a less restrictive constraint. Comparing the blue area and the orange area, we observe that in 6 out of the 9 tasks, the $\alpha$ for better inference performance is higher than the $\alpha$ that enables safe Q estimates, indicating that test-time inference requires milder constraints. In the next section, we show that with an MCEP, we can achieve much better inference performance without breaking the stable Q estimates.

### 5.3 Comparison on MuJoCo locomotion and Adroit

We compare the proposed method to state-of-the-art offline RL methods CQL and IQL, together with our baselines TD3BC and AWAC. Similar hyper-parameters are used for all tasks from the same domain. For our baseline methods (TD3BC and AWAC), we use the hyper-parameter recommended by their papers. TD3BC uses $\alpha = 2.5$ for its Q value normalizer and AWAC uses $1.0$ for the advantage value normalizer. In TD3BC-MCEP, the target policy uses $\tilde{\alpha} = 2.5$ and the MCEP uses $\alpha^e = 10$. In AWAC-MCEP, the target policy has $\tilde{\lambda} = 1.0$ and the MCEP has $\lambda^e = 0.6$. The full list of hyper-parameters can be found in the Appendix.

As is shown in Table 1, we observe that the evaluation policies with a mild constraint significantly outperform their corresponding target policy. TD3BC-MCEP gains progress on all *medium* and *medium-replay* datasets. Although the progress is superior, we observe a performance degradation on the *medium-expert* datasets which indicates an overly relaxed constraint for the evaluation policy. To overcome this imbalance problem, we designed a behavior-cloning normalizer. The results are shown in the Appendix. Nevertheless, the TD3BC-MCEP achieves much better general performance than the

target policy. In the AWAC-MCEP, we observe a consistent performance improvement over the target policy on most tasks. Additionally, evaluation policies from both TD3BC-MCEP and AWAC-MCEP outperform the CQL and IQL while the target policies have relatively low performances. On Adroit tasks, the best results are obtained by behavioral cloning agent and TD3BC with a high BC weighting factor. Other agents fail to outperform the BC agent. We observe that MCEP does not benefit these tasks where behavior cloning is essential for the evaluation performance.

## 5.4 Ablation Study

In this section, we design 2 groups of ablation studies to investigate the effect of the extra evaluation policy and its constraint strengths. Reported results are averaged on 5 runs of different random seeds.

**Performance of the extra evaluation policy.** Now, we investigate the performance of the introduced evaluation policy $\pi^e$. For TD3BC, we set the parameter $\alpha = \{2.5, 10.0\}$. A large $\alpha$ indicates a milder constraint. After that, we train TD3BC-MCEP with $\tilde{\alpha} = 2.5$ and $\alpha^e = 10.0$. For AWAC, we trained AWAC with the $\lambda = \{1.0, 0.5\}$ and AWAC-MCEP with $\tilde{\lambda} = 1.0$ and $\lambda^e = 0.5$.

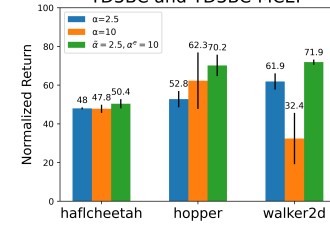 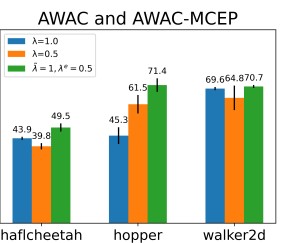

Figure 6: **Left**: TD3BC with $\alpha = 2.5$, $\alpha = 10$ and TD3BC-MCEP with $\tilde{\alpha} = 2.5, \alpha^e = 10$. **Right:** AWAC with $\lambda = 1.0$, $\lambda = 0.5$ and AWAC-MCEP with $\tilde{\lambda} = 1.0$ and $\lambda^e = 0.5$.

The results are shown in Figure 6. By comparing TD3BC of different $\alpha$ values, we found a milder constraint ($\alpha = 10.0$) brought performance improvement in hopper tasks but degrades the performance in walker2d tasks. The degradation is potentially caused by unstable value estimates (see experiment at section 5.2). Finally, the *evaluation policy* trained from the critic learned with a *target policy* with $\alpha = 2.5$ achieves the best performance in all three tasks. In AWAC, a lower $\lambda$ value brought policy improvement in hopper tasks but degrades performances in half-cheetah and walker2d tasks. Finally, an evaluation policy obtains the best performances in all tasks.

In conclusion, we observe consistent performance improvement brought by an extra MCEP that circumvents the tradeoff brought by the constraint.

**Constraint strengths of the evaluation policy.** We set up two groups of ablation experiments to investigate the performance of evaluation policy under different constraint strengths. For TD3BC-MCEP, we tune the constraint strength by setting the Q normalizer hyper-parameter. The target policy hyper-parameter is fixed to $\alpha = 2.5$. We pick three strengths for evaluation policy $\alpha^e = \{1.0, 2.5, 10.0\}$ to create more restrictive, similar, and milder constraints, respectively. For AWAC-MCEP, the target policy uses $\lambda = 1.0$. However, it is not straightforward to create a similar constraint for the eval-

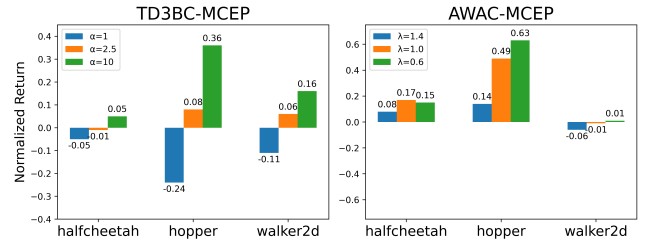

Figure 7: **Left:** TD3BC-EP with $\alpha = 1.0$, $\alpha = 2.5$ and $\alpha = 10.0$. **Right:** AWAC-EP with $\lambda = 1.4$, $\lambda = 1.0$ and $\lambda = 0.6$.

uation policy as it has a different policy improvement objective. We set $\lambda^e = \{0.6, 1.0, 1.4\}$ to show how performance changes with different constraint strengths.

The performance improvements over the target policy are shown in Fig. 7. The left column shows a significant performance drop when the evaluation policy has a more restrictive constraint ($\alpha^e = 1.0$) than the target policy. A very close performance is shown when the target policy and the evaluation policy have similar policy constraint strengths ($\alpha^e = 2.5$). Significant policy improvements are

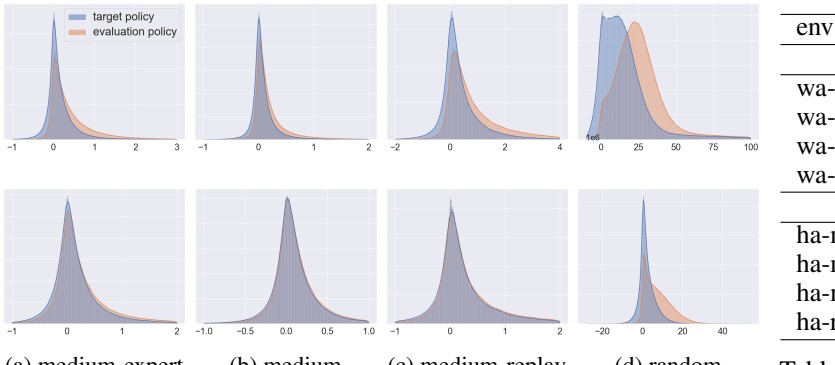

| env | $\tilde{\pi}$ (%) | $\pi^e$ (%) |
|---|---|---|
| TD3BC-MCEP | | |
| wa-me | 69.8 | 87.2 |
| wa-m | 66.2 | 82.7 |
| wa-mr | 71.8 | 88.7 |
| wa-r | 89.6 | 99.0 |
| AWAC-MCEP | | |
| ha-me | 63.4 | 70.8 |
| ha-m | 64.7 | 68.3 |
| ha-mr | 68.6 | 73.1 |
| ha-r | 75.3 | 95.6 |

(a) medium-expert    (b) medium    (c) medium-replay    (d) random

Figure 9: The distributions of $Q(s, \tilde{\pi}(s)) - Q(s,a)$ and $Q(s, \pi^e(s)) - Q(s,a)$ on MuJoCo locomotion tasks. **First row:** policies of TD3BC-MCEP learned in walker2d tasks. **Second row:** policies of AWAC-MCEP learned in half cheetah tasks. See the Appendix for full results.

Table 2: Proportion of $Q(s, \pi(s)) > Q(s,a)$ for target policies and evaluation policies in different tasks.

obtained with the target policy having a milder constraint ($\alpha^e = 10$). The right column presents the results of AWAC-MCEP. Generally, the performance in hopper tasks keeps increasing with milder constraints while the half-cheetah and walker2d tasks show performances that increase from $\lambda = 1.4$ to $\lambda = 1$ and similar performances between $\lambda = 1$ and $\lambda = 0.6$. Compared to the target policy, the evaluation policy consistently outperforms in half-cheetah and hopper tasks. On the walker2d task, a strong constraint ($\lambda = 1.4$) causes a performance worse than the target policy but milder constraints ($\lambda = \{1, 0.6\}$) obtain similar performance to the target policy.

In conclusion, for both algorithms, we observe that on evaluation policy, a milder constraint obtains higher performance than the target policy while a restrictive constraint may harm the performance.

### 5.5 Estimated Q values for the learned evaluation policies

To compare the performance of the policies learned in Section 5.3 on the learning objective (maximizing the Q values), we counted Q differences between the policy action and the data action $Q(s, \pi(s)) - Q(s,a)$ in the training data (visualized in Figure 9). Proportions of data points that show positive differences are listed in Table 2, where we find that on more than half of the data, both the target policy and the MCEP have larger Q estimation than the behavior actions. Additionally, the proportions for the MCEP are higher than the proportions for the target policy in all datasets, indicating that the MCEP is able to move further toward large Q values.

## 6 Conclusion

This work focuses on the policy constraints methods where the constraint addresses the tradeoff between stable value estimate and evaluation performance. While to what extent the constraint achieves the best results for each end of this tradeoff remains unknown, we first investigate the constraint strength range for a stable value estimate and for evaluation performance. Our findings indicate that test time inference requires milder constraints that can go beyond the range of stable value estimates. We propose to use an auxiliary *mildly constrained evaluation policy* to circumvent the above-mentioned tradeoff and derive a performant evaluation policy. The empirical results show that MCEP obtains significant performance improvement compared to target policy and achieves competitive results to state-of-the-art offline RL methods. Our ablation studies show that an auxiliary evaluation policy and a milder policy constraint are essential for the proposed method. Additional empirical analysis demonstrates higher estimated Q values are obtained by the MCEP.

**Limitations.** Although the MCEP is able to obtain a better performance, it depends on stable value estimation. Unstable value learning may crash both the target policy and the evaluation policy. While the target policy may recover its performance by iterative policy improvement and policy evaluation, we observe that the evaluation policy may fail to do so. Therefore, a restrictive constrained target policy that stabilizes the value learning is essential for the proposed method.

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
