# OpenReview forum: "Mildly Constrained Evaluation Policy for Offline Reinforcement Learning"
_NeurIPS.cc/2023/Conference — Submitted to NeurIPS 2023_

### Official Review · Reviewer_nJEQ · 2023-06-22

**Soundness:** 2 fair
**Presentation:** 2 fair
**Contribution:** 3 good
**Rating:** 6
**Confidence:** 4

**Summary:**

The paper investigates the problem of policy constraints in offline reinforcement learning (RL) settings, and finds the phenomenon that milder constraints on policies during training can lead to better performances at inference tests. The proposed component MCEP can be added on existing algorithms including TD3BC and AWAC. Experiments on D4RL dataset show improved performances over vanilla TD3BC and AWAC.

**Strengths:**

The strength of policy constraints in offline RL is an important problem. It is a novel perspective to separate the policies for value estimation and inference with different constraint levels. I would say the proposition that milder constraints can improve policy inference performances is an interesting problem. The experiments are thorough with necessary ablation studies, and the results indeed show some improvement by using the proposed constraining method.

**Weaknesses:**

I think one major critique of the paper is: the most essential discovery that milder constraints may be required for test-time inference is mostly from experimental evaluations. The observations are not even consistent for that only 6 out of 9 shows this pattern. This is not strong evidence showing that milder constraints are necessarily always better. Some theoretical analysis or at least insights about this observation can be provided to make it more convincing.

Another critique is that although the experiments show some improvement by using MCEP on TD3BC and AWAC and over some baselines like CQL and IQL. These are not the SOTA results on these offline datasets, there exists better algorithms proposed by the time of NeurIPS submission that should be aware of:

[1] Hansen-Estruch, Philippe, et al. "Idql: Implicit q-learning as an actor-critic method with diffusion policies." arXiv preprint arXiv:2304.10573 (2023).

[2] Garg, Divyansh, et al. "Extreme Q-Learning: MaxEnt RL without Entropy." arXiv preprint arXiv:2301.02328 (2023).

[3]  Wang, Zhendong, Jonathan J. Hunt, and Mingyuan Zhou. "Diffusion policies as an expressive policy class for offline reinforcement learning." arXiv preprint arXiv:2208.06193 (2022).

The paper writing can be further improved.

**Questions:**

How to evaluate the strength of restriction for Eq. (8) given the fact that it has an additional $Q$ term? I think it’s not fair to just say Eq. (8) is less restrictive than Eq. (6) since after taking an exponential function it is similar as Eq. (6) but with advantage $A$ replaced by $Q$.

For Fig. 5, I think it’s better to just give the $\alpha$ values in a table.

Fig. 4, are the results averaged across different seeds?

**Limitations:**

The limitations of the current method are discussed, that the evaluation policies in MCEP may not be consistent with the value function and can lead to unstable value estimation if used in policy evaluation.

---

> ### Author Rebuttal · Authors · 2023-08-09
>
> > I think one major critique of the paper ….
> >
>
> We appreciate reviewer nJEQ for their kind comments and review for our manuscript.
>
> We apologize for the misunderstanding of “not even consistent”, which is probably caused by our presentation. We argue that the experiment results of “6 out of 9” are **consistent** with our claim, even with a “5 out of 9” after we correct the mismatched axis (see Figure 3 in the submitted .pdf file)
>
> The “6 out of 9” (now 5 out of 9) comes from Section 5.2 and Figure 3 (pdf) refers to the number of tasks. Figure 3 (pdf) visualizes two areas. The orange area represents the constraint strengths that enable the evaluation policy of **TD3BC-MCEP** to outperform its target policy (which has a fixed constraint strength $\alpha=2.5$ for all tasks). In other words, in 7 out of 9 tasks, milder constraints enable the evaluation policy to outperform its target policy, which supports our claim.
>
> The blue area covers the constraint strengths of the actor in **TD3BC** that raise Q-value explosion during training. We visualize both areas in one figure to illustrate the benefit of wider policy space brought by MCEP (reaching the area beyond the blue area).
>
> Also, note that “Q does not explode” is not an indicator of a stable Q-estimate. i.e. In tasks where the orange area is covered by the blue area, unstable Q-estimate, or high Q-estimate error could still exist. We now introduce a new experiment to clearly illustrate it. We deploy a finer-grained hyperparameter search for TD3BC ($\alpha=\[1.0, 2.0, 2.5, 3.0, 4.0, … 10.0]$) and for TD3BC-MCEP ($\alpha^E=\[1.0, 2.0, 2.5, 3.0, 4.0, … 10.0]$ with a fixed $\alpha=2.5$ for the target policy). The goal is to find optimal constraint strengths for both methods in each task, as well as the corresponding inference performance.
>
> In Figure 1 Left (`.pdf`), we listed the corresponding constraint strengths ($\alpha^E$ for TD3BC-MCEP and $\alpha$ for TD3BC). In 'medium-replay" tasks, the $\alpha^E > \alpha$ indicates milder constraint improves performances. In "walker2d-medium-replay" task, the optimal $\alpha^E$ is found in the Q-explosion area which supports the discovery of Figure 3 (`.pdf`). In "medium" tasks, the optimal $\alpha^E == \alpha$ while the MCEP still outperforms TD3BC (see Figure 3 Right (`.pdf`) for the performance difference). This observation emphasizes that the Q-estimate error brought by milder constraints of the actor of TD3BC degrades the inference performance, even though these errors do not cause Q-value explosion. The insight here is milder constraints for the target policy will obtain larger Q-esimtate errors and these errors degrade the accuracy of the estimated Q function. The inaccurate Q function then misleads the optimization of the policy.
>
> In conclusion, in 8 of the 9 tasks, the optimal constraint strengths for TD3-MCEP are higher than its target policy ($\alpha=2.5$), and in 7 of the 9 tasks, the optimal policies found by TD3-MCEP outperforms optimal policies found by TD3. These results support our claim and confirm the effectiveness of the proposed MCEP approach.
>
> > Another critique is that although the experiments show some improvement by … there exists better algorithms proposed by the time of NeurIPS submission that should be aware of:
> [1] Hansen-Estruch, …
> [2] Garg, Divyansh, et al. …
> [3] Wang, Zhendong, …
> >
>
> Thanks for your kind suggestion. We are aware of the comparison to all the cited methods. To make fair comparisons, we consider the one-hyperparameter setting (same values for all tasks) and MuJoCo locomotion tasks. We rerun EQL[2] using the official implementation with the recommended hyperparameters. We rerun DQL[3] by sweeping the constraint $\eta=[1.0, 2.0, 2.5]$ (following the sweeping strategy in [1]) and found that the paper-recommended $\eta=1.0$ performs the best. Due to the time limit, we did not rerun IDQL [1] so we use the reported results. Among these methods, DQL performs the best. Full results show in the table of General Rebuttal (unable to post in this thread due to word limits). We found that our methods TD3BC-MCEP and AWAC-MCEP outperform [1] and [2]. [3] shows superior results among all methods.
>
> In addition, we integrated the proposed MCEP to the most performant [3], using the constraint hyper-parameter $\tilde{\eta} = 1.0$ and using a milder constraint strength $\eta^e=2.5$ for the evaluation policy. The resulting DQL-MCEP obtains improved performance based on DQL, which further verifies the effectivenesses of the proposed approach.
>
> > The paper writing can be further improved.
> >
>
> All authors promise to carefully proofread this paper for improving the writing.
>
> > How to evaluate the strength of restriction for Eq. (8) given the fact that it has an additional Q term? I think it’s not fair to just say Eq. (8) is less restrictive than Eq. (6) since after taking an exponential function it is similar as Eq. (6) but with advantage A replaced by Q.
> >
>
> Thanks for your kind comments. To make a fair comparison of constraint strengths, we re-design the Eq. (8) by replacing the Q term with advantage A. Using the advantage still obtains significant performance improvement based on the AWAC. The results are shown in the table above.
>
> $$
> \mathcal{L}_{\pi^e} (\phi) = \mathbb{E}_{s, a \sim \mathcal{D}, \hat{a} \sim \pi^e (\cdot|s) }\[-A(s, \hat{a}] - \lambda \log \pi^e_{\phi} (a|s)\]
> $$
>
>
> > For Fig. 5, ...
> >
>
> The new experiment in Figure 3 Left (`.pdf`) makes a clear comparison to the $\alpha$ values. We will also provide tables in the Appendix.
>
> > Fig. 4, … across different seeds?
> >
>
> TD3BC (left column and middle column) are single-seed results while AWAC is averaged among 5 seeds. We will replace this figure with Figure 2 (`.pdf`) where all curves are averaged among 5 seeds.

---

> > ### Comment · Reviewer_nJEQ · 2023-08-16
> > **Response to rebuttal**
> >
> > I would like to thanks the authors for providing rebuttals with additional experiments.
> >
> > For sentence "in 8 of the 9 tasks, the optimal constraint strengths for TD3-MCEP are higher than its target policy", how to see 8 tasks show higher constraint values? Also, should it be milder constraint strengths but higher values?  Overall, this value to strength mapping is a bit odd, why not using a $1/\alpha$ value to directly indicate the constraint strength?
> >
> > The other remaining weakness of the paper is: 1. lack of theoretical insight and supports; 2. writing of the paper, which I hope will be improved.
> >
> > It's impressive to see this method is also applied on DQL and improves its performance. Thanks for adding these results.

---

> > > ### Author Response · Authors · 2023-08-16
> > > **Reply to Reviewer nJEQ**
> > >
> > > Thank you for your reply.
> > > > For sentence "in 8 of the 9 tasks, the optimal constraint strengths for TD3-MCEP are higher than its target policy", how to see 8 tasks show higher constraint values?
> > >
> > > In Left of Figure 1 (submitted `.pdf`), there are 8 orange triangles having $\alpha^E>2.5$ and their target policies have $\tilde{\alpha}=2.5$ (not visualized). Hence we say they have higher $\alpha$ values than the target policies, indicating that they have milder constraint strengths than target policies.
> > >
> > > > Also, should it be milder constraint strengths but higher values?
> > >
> > > Yes, higher $\alpha$ values indicate milder constraint strengths.
> > >
> > > > Overall, this value to strength mapping is a bit odd, why not use a $\frac{1}{\alpha}$ value to directly indicate the constraint strength?
> > >
> > > We agree that representing with $\frac{1}{\alpha}$ is more clear. We will replace the y-axis with $\frac{1}{\alpha}$ values.
> > >
> > > > 1. lack of theoretical insight and supports;
> > >
> > > While we did not provide formal theoretical results, we think the intuition behind the empirical findings is clear: OOD constraints can reduce value estimation error while restraining the policy search space. The target policy needs restrictive constraints to reduce value-estimate errors that could be amplified by bootstrapping. An evaluation policy searches policies without influencing value estimate, its milder constraint provides a larger policy space for searching performant policy.
> > >
> > > > 2. writing of the paper, which I hope will be improved.
> > >
> > > Appreciated for the comment. We will try our best to improve the writing in the next version of the paper.

---

> > > > ### Comment · Reviewer_nJEQ · 2023-08-19
> > > > **Response to author**
> > > >
> > > > Thanks for clarifying these points. I have no further questions for now. I will raise my score to weak acceptance after review.

---

### Official Review · Reviewer_FqHz · 2023-07-03

**Soundness:** 2 fair
**Presentation:** 2 fair
**Contribution:** 1 poor
**Rating:** 3
**Confidence:** 4

**Summary:**

This paper proposes Mildly Constrained Evaluation Policy (MCEP) for offline reinforcement learning to address the issue of excessively restrictive constraints for action selection during test time inference. MCEP uses a more constrained target policy for value estimation and another less restrictive policy for performance evaluation. Empirical results demonstrate the effectiveness of MCEP.

**Strengths:**

MCEP is easy to implement and can be plugged into many policy constraint offline RL methods.

**Weaknesses:**

Since $\pi_e$ does not participate in the policy evaluation, I think line 7 of Algorithm 1 can be removed and $\pi_e$ can be extracted from Q after actor critic learning to save computational cost. The contribution of MCEP is only to extract a less restrictive policy after RL learning, which is somewhat limited.

The overall idea of the paper is quite simple. However, the notations and descriptions are a bit confusing. For example, the notations in Algorithm 1 lack a clear definition ($\psi, \phi, \tilde \pi, \pi^e, \tilde w, w^e, \mathcal L(.,.)$). And $\psi$ and $\phi$ in line 6 and 7 of Algorithm 1 are reversed, since Q evaluation in Equation 2 is associated with $\phi$.

No theory supports MCEP in the paper.

**Questions:**

I do not have additional questions.

**Limitations:**

More hyperparameters need to be tuned for MCEP compared with the original algorithm.

---

> ### Author Rebuttal · Authors · 2023-08-09
>
> We thank the reviewer FqHz for their feedback and comments of our manuscript.
>
> > Since $\pi_e$ does not participate in the policy evaluation, I think line 7 of Algorithm 1 can be removed and $\pi_e$ can be extracted from Q after actor critic learning to save computational cost. The contribution of MCEP is only to extract a less restrictive policy after RL learning, which is somewhat limited.
> >
>
> We thank the reviewer for the suggestion of extracting the evaluation policy after training the critic network to optimize the computation cost. Different approaches for evaluation policy optimization may influence efficiency. However, in our experiments, we found that iterative updating is stable for learning the evaluation policy and the significant performance improvements confirm the stability.
>
> This work investigates the role of policy constraints in policy constraints offline RL methods. This is an important fundamental problem for offline RL research. Furthermore, our empirical analysis provides insights into constraint strengths for stable Q estimate and inference-time performance. This insight explains the mediocre performance of policy constraint methods. The proposed MCEP differs from existing approaches as it circumvents the solving of the trade-off between stable Q estimates and test-time inference. MCEP enables mitigating the Q estimate error and achieving milder constraints for better inference performance at the same time.
>
> MCEP is a simple yet effective and general approach for offline RL. It enables conventional policy constraint methods (e.g. TD3BC and AWAC) to achieve SOTA-level performances and enables SOTA policy constraint methods (e.g. Diffusion-QL) to obtain further performance improvements. The generality, simplicity, and strong empirical performance are the main strengths of our paper.
>
> > The overall idea of the paper is quite simple. However, the notations and descriptions are a bit confusing. For example, the notations in Algorithm 1 lack a clear definition ($\psi, \phi, \tilde{\pi}, \pi^e, \tilde{w}, w^e, \mathcal{L}(.,.)$).
> And $\psi$ and $\phi$ in line 6 and 7 of Algorithm 1 are reversed, since Q evaluation in Equation 2 is associated with $\phi$.
> >
>
> 1. We will carefully revise some contents to make the notation definition more clear. We use these notations to distinguish different components of the proposed approach. As introduced in Section 4.1, for the proposed MCEP, $\psi$ and $\phi$ are parameters of policies to optimize (e.g. neural network weights). $\psi$ corresponds to the target policy $\tilde{\pi}$ (i.e. the actor in actor-critic). $\phi$ corresponds to evaluation policy $\pi^E$ that the algorithm returns. $\tilde{w}$ and $w^E$ are policy constraint hyper-parameters w.r.t target policy and evaluation policy. $\mathcal{L}(.., ..)$ is a notation of the Loss function and it is widely used in RL and ML papers. We hope that Figure 1 would help illustrate these components and their notations.
> 2. Equation 2 is shown in the Background section, the introduction to policy constraint methods. As we mentioned above, in MCEP, $\phi$ refers to an evaluation policy that the algorithm returns. Therefore we use $\phi$ in line 7 of Algorithm 1. We use $\phi$ in Equation 2 as this policy is also returned by the algorithm.
>
> In policy constraint methods, the target policy and evaluation policy refers to the same policy. This dual identity of this actor actually motivates the proposed approach to separate the actor into a target policy and an evaluation policy. We will add content to detail the notation definitions and to improve the clarifications.
>
> > No theory supports MCEP in the paper.
> >
>
> This work provides empirical analysis and insights instead of providing theoretical analysis. We provide a range of experimental results that
>
> 1) show the problems of overly restrictive constraints for the target policy (actor),
>
> 2) reveal the relation between constraint strengths for stable Q estimate and inference-time performance,
>
> 3) implements instances of the proposed general approach to convention and SOTA policy constraints methods and makes fair comparisons to SOTA offline RL methods
>
> 4) and verify the effectiveness of the milder constraints and the extra evaluation policy in our ablation study
>
> The theory behind the empirical analysis of this work is an interesting direction to explore.

---

> > ### Comment · Area_Chair_1Teu · 2023-08-19
> >
> > Thanks for your rebuttal, it has been noted.

---

### Official Review · Reviewer_sDxq · 2023-07-05

**Soundness:** 4 excellent
**Presentation:** 3 good
**Contribution:** 3 good
**Rating:** 6
**Confidence:** 3

**Summary:**

This work addresses the issue of excessive policy constraints in stabilizing value estimation within the offline RL paradigm. A separate target policy is used solely for evaluation and stabilizing value estimation, which is more constrained than the "evaluation policy." The evaluation policy does not participate in policy evaluation and is improved by the value function estimates, with the level of constraint adjusted by the weight of the term.

**Strengths:**

Major points:
- This procedure can be easily integrated into offline RL algorithms that utilize policy constraints, and empirical results apply it to TD3+BC and AWAC. The empirical findings demonstrate promising improvements, with baselines encompassing the standard suite of state-of-the-art offline RL algorithms.
- The paper is well-written, easy to comprehend, and thoughtfully structured.
- The idea itself is intuitive, and the toy experiments convincingly demonstrate that over-constraint poses a significant issue. Figure 2 clearly illustrates the adverse effects of over-constraint, with the policy performing poorly in low state value regions of the maze.
- The ablation studies are extensive and demonstrate the method's effectiveness.

I believe this simple yet intuitive method is worth presenting to the broader offline RL community. I believe this work should be accepted.

**Weaknesses:**

Major points:
- While the results show promise, they do not indicate substantial improvements across many environments, and there is some inconsistency observed. The method shows a decrease in performance in the medium-expert D4RL tasks and the pen task.

**Questions:**

Can you explain the inconsistencies in the results, especially in the pen task? What could be a possible reason?

**Limitations:**

Yes

---

> ### Author Rebuttal · Authors · 2023-08-09
>
> We appreciate reviewer sDxq for their review and kind comments for our manuscript.
>
> > While the results show promise, they do not indicate substantial improvements across many environments, and there is some inconsistency observed. The method shows a decrease in performance in the medium-expert D4RL tasks and the pen task.
> Can you explain the inconsistencies in the results, especially in the pen task? What could be a possible reason?
> >
>
> In our experiment results.
>
> 1. TD3BC-MCEP shows weaker performances than TD3BC on “halfcheetah-m-e” and “hopper-m-e” tasks.
> 2. The results of -MCEP is slightly weaker than their baseline (TD3BC and AWAC) in Adriot tasks.
>
> For all the abovementioned cases, we find that behavior cloning loss is the main contributor to the final performances. To show this, we present the results of a Top-10% behavior cloning agent (behavior cloning using 10% data of higher return).
>
> | Dataset      | Top10 BC | TD3BC-MCEP | AWAC-MCEP | EQL | IDQL | DQL | DQL-MCEP |
> | ------------ | -------- | ---------- | --------- | --- | ---- | --- | -------- |
> | hafcheetah-m |$43.1\pm0.3$ | $55.5\pm0.4$|$46.9\pm0.0$|$46.5\pm0.1$ | $49.7$  | $49.8\pm0.2$  | $53.2\pm0.2$ |
> | hopper-m     |$56.9\pm1.6$ |$91.8\pm0.9$|$98.1\pm0.6$| $67\pm1.3$  |  $63.1$    |$81.7\pm6.6$  | $95.5\pm2.2$  |
> | walker2d-m   |$73.3\pm2.5$  |$88.8\pm0.5$|$81.4\pm1.6$| $81.8\pm1.1$ | $80.2$     |$85.5\pm0.8$  | $75.3\pm3.6$ |
> | hafcheetah-mr |$39.9\pm0.8$ |$50.6\pm0.2$|$44.9\pm0.1$|  $43.1\pm0.5$ | $45.1$     |$47\pm0.2$  | $47.8\pm0.1$ |
> | hopper-mr     |$72\pm2.1$   |$100.9\pm0.4$|$101.1\pm0.2$| $97.3\pm3.3$| $82.4$     |$100.6\pm0.2$  | $100.9\pm0.1$  |
> | walker2d-mr   |$56.6\pm3.3$ |$86.3\pm3.2$|$83.4\pm0.8$|  $71.4\pm4.7$ | $79.8$     |$93.6\pm2.5$   | $92.6\pm2.1$   |
> | hafcheetah-me |$93.5\pm0$  |$71.5\pm3.7$|$69.5\pm3.8$|   $89.4\pm1.6$ |  $94.4$    |$95.7\pm0.4$  | $93.4\pm0.8$   |
> | hopper-me     |$108.9\pm0.0$  |$80.1\pm12.7$|$84.3\pm16.4$|  $97.3\pm3.3$| $105.3$ | $102.1\pm3.0$  |$107.7\pm1.5$  |
> | walker2d-me   |$111.1\pm0.5$  |$111.7\pm0.3$|$110.1\pm0.2$|  $109.8\pm0.0$| $111.6$     | $109.5\pm0.1$  | $109.7\pm0.0$   |
> | Average      |$72.8$   | $81.9$    | $79.9$  |  $78.1$   | $79.0$  | $85.0$ |  $86.2$   |
>
> We observe that Top-10% BC agent shows superior performances on “medium-expert” tasks. These results are higher or similar to RL methods. In Adroit tasks, Behavior cloning also shows superior performance, as well as the TD3BC with a high coefficient for the BC loss (see Table 1 in the paper).
>
> In these tasks, high-quality data (e.g. expert data) exists and optimal actions can be inferred within the data distribution. As analyzed in [1-3], the estimated Q values for OOD actions could diverge and becomes arbitrarily high (may be much higher than the accurate estimate of optimal actions inside the dataset). In this case, a mild policy constraint could let the policy exploit these high but incorrect Q values so resulting in bad-quality evaluation policies.
>
> In other datasets, expert data does not exist in the dataset and the policy is required to improve **over** the dataset. Therefore, the problems of estimating the value of the state-action pairs (maybe not in the datasets) and exploring the critic network become important. In other words, balancing the tradeoff between stable Q estimate and test-time inference is key to obtaining performant policies. The proposed approach is effective for these tasks as the overly constrained target policy mitigates the Q-estimate error and the MCEP achieves better test-time inference.
>
> [1] Fujimoto, S., Meger, D. and Precup, D., 2019, May. Off-policy deep reinforcement learning without exploration. In International conference on machine learning (pp. 2052-2062). PMLR.
>
> [2] Kumar, A., Fu, J., Soh, M., Tucker, G. and Levine, S., 2019. Stabilizing off-policy q-learning via bootstrapping error reduction. Advances in Neural Information Processing Systems, 32.
>
> [3] Kumar, A., Zhou, A., Tucker, G. and Levine, S., 2020. Conservative q-learning for offline reinforcement learning. Advances in Neural Information Processing Systems, 33, pp.1179-1191.

---

> > ### Comment · Reviewer_sDxq · 2023-08-17
> > **Rebuttal response**
> >
> > I thank the author's for the new results.
> >
> > In light of the theoretical concerns also mentioned by other reviewers, I keep my original rating of leaning towards acceptance.

---

### Official Review · Reviewer_wQCk · 2023-07-07

**Soundness:** 3 good
**Presentation:** 4 excellent
**Contribution:** 2 fair
**Rating:** 6
**Confidence:** 5

**Summary:**

Offline reinforcement learning (RL) methods frequently involve a policy constraint to mitigate error propagation when learning the Q function. Generally, a single constraint strength is used throughout training. This paper proposes instead to use different constraint strengths for learning the target policy, which is only used for learning the Q function, and the evaluation policy, which is the final policy returned by the algorithm. In particular, a stronger constraint is needed to ensure stability when training the target policy, but weakening the constraint for the evaluation policy can lead to better performance.

**Strengths:**

* The idea is fairly general and can be instantiated with various RL algorithms, as shown in the paper.
* The experimental results provide insight into the role of the constraint and the tradeoff between stability and performance.
* Conceptually, the approach allows for a continuum of algorithms between one-step RL and standard actor-critic methods.
* The paper is clearly written and understandable.

**Weaknesses:**

* The algorithm introduces an additional hyperparameter that requires tuning, which is already a challenge in offline RL.
* The paper found that “in 6 out of the 9 tasks, the $\alpha$ for better inference performance is higher than the $\alpha$ that enables safe Q estimates”. While 6/9 is a majority, this is not convincing evidence that weakening the constraints is always helpful.

**Questions:**

In Section 5.2, how exactly do you determine "the lowest $\alpha$ value that causes Q value explosion"? (In particular, how is "explosion" defined?) Could this lead to a hyperparameter tuning procedure that involves only looking at the Q values and does not require off-policy evaluation or sample collection?

**Limitations:**

Yes, limitations are addressed.

---

> ### Author Rebuttal · Authors · 2023-08-09
>
> We appreciate reviewer wQCk for their review of our manuscript.
>
> > The algorithm introduces an additional hyperparameter that requires tuning, which is already a challenge in offline RL.
> >
>
> Compared to policy constraints methods, the proposed MCEP method has an extra hyperparameter for the constraint strength of evaluation policy. Separate constraint strengths for the evaluation policy and the target policy are important to find optimal values respectively for stabilizing Q-estimate and obtaining better inference performance.
>
> In practice, we found that the performance is not sensitive to the value of this extra hyperparameter. E.g. The orange region in Figure 5 indicates values of this hyperparameter that enable the evaluation policy to outperform the target policy, which widely covers the hyperparameter space.
>
> In addition, we use a simple hyperparameter search strategy and find it works effectively. We use paper-recommended constraint strengths for the target policy and tune the strengths for the evaluation policy to milder strengths.
>
> > The paper found that “in 6 out of the 9 tasks, the $\alpha$ for better inference performance is higher than the  $\alpha$ that enables safe Q estimates”. While 6/9 is a majority, this is not convincing evidence that weakening the constraints is always helpful.
> >
>
> Figure 5 shows that in 6 out of the 9 tasks (5 out of the 9 tasks in the corrected version, Figure 3 of the submitted `.pdf` file), policy constraint strengths that enable evaluation policy to outperform target policy (the target policy has a fixed $\alpha=2.5$) may fall in the strengths that the same values will cause Q-explosion if assigned to the target policy. The results of these 5 tasks present one respect of unstable Q-estimate, i.e. high Q-estimate error that causes Q-value explosion. In the remaining $9-5=4$ tasks, where the Q-value does not explode, the evaluation policy with milder constraints still outperforms its target policy in 2 of the 4 tasks, in total, 7 out of the 9 tasks support our claim, which is consistent.
>
> We also provide an additional experiment (see Figure 1 `.pdf`) that investigates optimal policy strengths of TD3BC-MCEP and TD3BC. The results provide further insights that Q-estimate errors brought by mildly constrained target policy may still degrade the inference performance even though the Q-value does not explode.
>
> > In Section 5.2, how exactly do you determine "the lowest alpha value that causes Q value explosion"? (In particular, how is "explosion" defined?) Could this lead to a hyperparameter tuning procedure that involves only looking at the Q values and does not require off-policy evaluation or sample collection?
> >
>
> The blue area indicates $\alpha$ values of the TD3BC method. Under each $\alpha$ value, the training is run with  5 seeds and the Q value $Q(s, \pi(s))$ during training are visualized. If any one of these 5 runs shows Q-value explosion (Q value diverges and the policy performance is largely degraded), we consider this $\alpha$ value raises explosion and it will not be included in the blue area. Finally, we take the lowest value from those raised Q explosions as edges of the blue area.
>
> > Could this lead to a hyperparameter tuning procedure that involves only looking at the Q values and does not require off-policy evaluation or sample collection?
> >
>
> To investigate this approach, we present the full results of the hyperparameter searching in Figure 1 in the submitted .pdf file. In the case of TD3BC, we found that the optimal strengths are not always the milder ones without raising the Q explosion. Milder constraint introduces large Q-estimate errors and harms the inference-time performance, even though the Q values are not exploded (Q values explode when this error is high enough).

---

> > ### Comment · Reviewer_wQCk · 2023-08-21
> >
> > Thank you for answering my questions and providing additional experimental results. My concerns are largely addressed, and I am still in favor of acceptance. While the proposed method is simple, simplicity is not inherently bad IMO – indeed, simplicity has benefits as well – and the fact the idea works with several algorithms is evidence that it is general and will be useful to the research community.

---

### Author Rebuttal · Authors · 2023-08-09

## Summary of Positive Feedback
We would like to thank the reviewers for their thorough reviews and detailed feedback. It appears reviewers have perceived various aspects of this work's contribution.
- reviewer nJEQ commented the studied problem *"is an important problem"* and the proposition of our main claim is "an interesting problem"
- Reviewer wQCk commented that our approach is *"fairly general"* and *" provides insight into the role of the constraint"*. Reviewer sDxq commented that *"The idea itself is intuitive"* and *"believe this simple yet intuitive method is worth presenting to the broader offline RL community"*.
-  reviewer wQCk commented on the paper *"clear written"* and reviewer sDxq commented *"well-written, easy to comprehend, and thoughtfully structured."*.

In addition, we are glad to see all the reviewers agreed that the proposed approach is simple yet general.

## Novelty
**1. Methodology:** We present Mildly Constrained Evaluation Policy (MCEP) deriving an extra policy from the critic for evaluation, with an overly constrained target policy to mitigate the Q-estimate error by OOD actions. Our approach circumvents a well-known trade-off in policy constraint offline RL methods: stable Q-estimate and policy inference performance.

**2. Conceptually:** We study the roles of the policy constraint: stabilize Q estimate and test-time inference. Our approach separates the roles into two policies: a target policy and an evaluation policy.

**3. Empirically:** Despite the simplicity, the proposed general method enables conventional policy constraint methods (TD3BC and AWAC) to achieve SOTA-level performances and enables SOTA policy constraint methods (Diffusion-QL) to obtain further performance improvements.


## Empirical analysis
1. We argue that the experiment results shown in Figure 5 from Section 5.2 are **consistent** with our claim of "milder policy constraint is required for test-time inference than the constraint of stable Q-estimate". Reviewer wQCk and Reviewer nJEQ commented *"Results in 6 out of 9 environments support our claim but others do not"*. The 6 tasks point to those constraint strengths that may raise Q-value explosion if assigned to the target policy. But in 7 out of 9 environments (the range of the orange area is larger than 0 on 7 axes), a wide range of milder constraint strengths enable the evaluation policy to outperform the target policy.

2. To further clarify its insight, we introduce a fine-grained hyper-parameter search to compare the optimal constraint strengths for evaluation policy/actor (Figure 1 `.pdf`). In 8 out of 9 tasks, the evaluation policy finds its optimal constraint milder than its target policy. The results also show the performance degradation caused by Q-estimate error even when the Q-value does not explode.

3. As commented by reviewer nJEQ *"there exists better algorithms"*, we update our performance evaluation by introducing the following agents:
    1) A behavior cloning agent with $10\%$ highest-return data (Top10BC).
    2) comparison to 3 SOTA methods mentioned by reviewer nJEQ (EQL, IDQL, DQL).
    3) DQL-MCEP by applying the proposed MCEP to a SOTA policy constraints method DQL (DQL-MCEP).

| Dataset      | Top10 BC | TD3BC-MCEP | AWAC-MCEP | EQL | IDQL | DQL | DQL-MCEP |
| ------------ | -------- | ---------- | --------- | --- | ---- | --- | -------- |
| hafcheetah-m |$43.1\pm0.3$ | $55.5\pm0.4$|$46.9\pm0.0$|$46.5\pm0.1$ | $49.7$  | $49.8\pm0.2$  | $53.2\pm0.2$ |
| hopper-m     |$56.9\pm1.6$ |$91.8\pm0.9$|$98.1\pm0.6$| $67\pm1.3$  |  $63.1$    |$81.7\pm6.6$  | $95.5\pm2.2$  |
| walker2d-m   |$73.3\pm2.5$  |$88.8\pm0.5$|$81.4\pm1.6$| $81.8\pm1.1$ | $80.2$     |$85.5\pm0.8$  | $75.3\pm3.6$ |
| hafcheetah-mr |$39.9\pm0.8$ |$50.6\pm0.2$|$44.9\pm0.1$|  $43.1\pm0.5$ | $45.1$     |$47\pm0.2$  | $47.8\pm0.1$ |
| hopper-mr     |$72\pm2.1$   |$100.9\pm0.4$|$101.1\pm0.2$| $97.3\pm3.3$| $82.4$     |$100.6\pm0.2$  | $100.9\pm0.1$  |
| walker2d-mr   |$56.6\pm3.3$ |$86.3\pm3.2$|$83.4\pm0.8$|  $71.4\pm4.7$ | $79.8$     |$93.6\pm2.5$   | $92.6\pm2.1$   |
| hafcheetah-me |$93.5\pm0$  |$71.5\pm3.7$|$69.5\pm3.8$|   $89.4\pm1.6$ |  $94.4$    |$95.7\pm0.4$  | $93.4\pm0.8$   |
| hopper-me     |$108.9\pm0.0$  |$80.1\pm12.7$|$84.3\pm16.4$|  $97.3\pm3.3$| $105.3$ | $102.1\pm3.0$  |$107.7\pm1.5$  |
| walker2d-me   |$111.1\pm0.5$  |$111.7\pm0.3$|$110.1\pm0.2$|  $109.8\pm0.0$| $111.6$     | $109.5\pm0.1$  | $109.7\pm0.0$   |
| Average      |$72.8$   | $81.9$    | $79.9$  |  $78.1$   | $79.0$  | $85.0$ |  $86.2$   |

A minor issue fixed:
We correct the task-name mismatch problem for the orange area in Figure 5, resulting in Figure 3 in the submitted .pdf file.

We will address individual comments from the reviewers by replying to separate threads below.

---

### Decision · Program_Chairs · 2023-09-21

**Decision:**

Reject

**Comment:**

I've gone through the discussions and paper in detail. I found the algorithm interesting and the paper well-written, as did most of the reviewers. However, I tend to agree with the main concern of most reviewers regarding the limited significance of the contribution. I think the easiest way for the authors to improve on this is to extend the experiments to more varied and complex domains. I encourage the authors to look to domains like robotics, game-playing, web navigation, drone/navigation/driving, operations, and see if one or more of these can offer more interesting datasets and environments on which they can expand their empirical analysis.